# Effects of Open-Skill Exercises on Cognition on Community Dwelling Older Adults: Protocol of a Randomized Controlled Trial

**DOI:** 10.3390/brainsci11050609

**Published:** 2021-05-10

**Authors:** Wei Guo, Biye Wang, Małgorzata Smoter, Jun Yan

**Affiliations:** 1College of Physical Education, Yangzhou University, Yangzhou 225000, China; guowei@yzu.edu.cn (W.G.); wangbiye@yzu.edu.cn (B.W.); 2Institute of Sports, Exercise and Brain, Yangzhou University, Yangzhou 225000, China; 3Gdańsk Academy of Physical Education and Sport, 80-001 Gdańsk, Poland; m.h.smoter@gmail.com

**Keywords:** open-skill exercises, closed-skill exercises, cognition, older adults

## Abstract

(1) Cognitive function may benefit from physical exercise in older adults. However, controversy remains over which mode of exercise is more beneficial. (2) The aim of the proposed study is to investigate the effect of open-skill exercise training on cognitive function in community dwelling older adults compared with closed-skill exercise, cognitive training, and active control. (3) One hundred and sixty participants, aged between 60 and 80 years old, will be recruited from community senior centers in Yangzhou, China and randomly assigned to one of four groups: open-skill exercise group, closed-skill exercise group, mobile game playing group, and active control group. All participants will join a 24-week program involving 50 min sessions three times a week. The primary outcome measure is visuospatial working memory. Secondary measures include subjective memory complaint, attention network, nonverbal reasoning ability, and physical activities. All participants will be measured before, mid-way, and immediately after intervention, and three months later. (4) If successful, this study is expected to provide evidence-based recommendations for older adults to select the most efficient and effective mode of exercise to improve cognitive function. Importantly, the three intervention groups provide an opportunity to separate the cognitive activity component from the physical activity component. Comparison of these components is expected to help elucidate possible mechanisms contributing to the additional cognitive benefit of open-skill exercises.

## 1. Introduction

Data issued by the National Bureau of Statistics of China indicate that there were 249 million (represent 17.9% of the total) people aged over 60 and 167 million (represent 11.9% of the total) aged over 65 in China in 2019. The prevalence of dementia in elderly people over 65 years old is 3.2–9.9% [1], which means that there are 5.3–16.5 million dementia patients in China. Cognitive function declines with aging, especially in people with neurodegenerative diseases, such as dementia and Alzheimer’s [2]. Aging-related cognitive decline and cognitive impairment impact the quality and expectancy of life in old people [3], and is of great public health and economic concern. Of note, certain types of cognitive variables remain intact, while others become impaired, with cognitive aging. Salthouse et al., (2010) demonstrated a nearly linear decline from early adulthood with respect to memory, spatial visualization, speed, and reasoning, whereas vocabulary knowledge could still increase until the age of 60 [4]. That is to say, “held” cognitive variables include vocabulary and general information in which the relevant acquisition occurred earlier in one’s life, while the “non-held” variables tend to involve the manipulation and transformation of information. Thus, the proposed study will mainly focus on aspect of cognition that decline with aging.

Cognitive function reacts sensitively to environmental changes, and can be significantly improved after certain intervention. Several studies have shown that cognitive function may benefit from physical exercise in older adults [5,6,7]. Yet, Diamond et al., (2016) argued that aerobic exercise without any cognitive component produces little or no cognitive benefit. The authors predicted that, besides direct exercise training, successful approaches should incorporate cognitive, emotional, and social needs [8]. The authors also stressed that certain modes of physical exercise, including aerobic-exercise and resistance-training, are among the least effective ways of improving executive functions [9]. Therefore, physical training should extend beyond simple moving to moving with thought [10]. Fortunately, various studies have started to focus on identifying cognitive benefit from combinations of cognitive training and physical exercise. Some studies examined these components separately, while others conducted the two components simultaneously. The separate approach usually involves cognitive trainings (e.g., focus on memory, attention, executive function, visuospatial ability), followed by physical exercise (e.g., aerobic exercise, resistance training), or vice versa [11,12,13,14,15,16]. The simultaneous approach is usually conducted in dual-tasks, such as completing the cognitive task during physical exercise [17,18,19,20] or in exergames, which combine physical exercise with computer-simulated environments and interactive videogame features [7,21]. In general, that simultaneous approach tends to have a larger effect size than the separate approach on cognition [22,23]. However, concerns exist that the similarity between cognitive assessments and dual-tasks produce a learning effect that impacts the reliability of the results. Laboratory based intervention cannot easily be generalized to everyday life of the elderly.

Thus, the present study introduces a new approach with high ecological validity by combining physical exercise and cognitive training simultaneously, specifically open-skill exercises. Open-skill exercises are defined as exercises in which participants are required to react in a dynamically changing, unpredictable, and externally environment (e.g., table tennis, badminton). In contrast, closed-skill exercises are conducted in an environment that is relatively stable, predictable, and self-paced (e.g., running and swimming) [24]. Previous studies confirmed that older adults demonstrated better executive function when participating in open-skill exercises compared to closed-skill exercises [25,26,27,28,29]. Children who played open-skill exercises also exhibited better cognitive functions and academic achievement than those who played closed-skill exercises [28,30,31]. Open-skill exercises might be advantageous because participants must mentally compare the present situation with past ones, and use this information to predict likely outcomes; consequently, participants must use their cognitive functions.

Our previous cross-sectional study on healthy older adults showed that both closed-skill and open-skill exercisers exhibited better visuospatial working memory compared to sedentary older adults. Specifically, open-skill exercises that demand higher cognitive processing exhibited selective benefits for the passive maintenance of working memory [32]. However, the cross-sectional design could only reveal a possible relationship, not a causal relationship. Additionally, the reason why open-skill exercise produces additional cognitive benefits to working memory remains unknown. Therefore, the present study protocol provides a first step to initiate an intervention study to answer the question that which type of exercise is better for older adults to decay cognitive decline. The protocol will further reveal the mechanism by setting a cognitive training group besides the open-skill and closed-skill exercise groups. Open-skill exercises can be viewed as a combination of physical exercise and cognitive training, closed-skill exercises and cognitive trainings can be viewed as pure physical exercise or cognitive training. This design may provide the opportunity to separate the physical activity component and cognitive activity component by comparing differences among the groups.

The primary objective is to investigate the effect of 24 weeks of open-skill exercise training on the cognitive function of heathy older adults in comparison to the closed-skill exercise group, cognitive training group, and control group. It is hypothesized that both training regimes will have a positive effect on cognitive function compared to the control, with the largest effect occurring in the open-skill exercise group.

## 2. Materials and Methods

### 2.1. Study Design

This study is a double-blind, 24-week randomized controlled trail (RCT) with three experimental intervention groups and an active control group. Participants will be randomly allocated to one of the four groups. The outcome measures will be performed at baseline, mid-way through intervention, immediately after intervention, and at a 3-month post-intervention follow-up. All tests and measures will be conducted by trained professionals who are blinded to the group assignment. The study design is presented in Figure 1. The schedule of enrolment, interventions, and assessments is presented in Figure 2.

The study protocol has been approved by the Ethics Committee of Yangzhou University (YXYLL-2020-106). The methods for consent were registered at Chinese Clinical Trail Registry (ChiCTR2000038733), and the data will be uploaded at the site when finishing the trial. The informed consent form will be obtained from all participants.

### 2.2. Participant Recruitment and Selection

Participants aged between 60 and 80 years will be recruited from the community senior centers in Yangzhou, China. Both male and female participants will be used. The older adults who are interested in participating in the research will be subject to a short interview to confirm their eligibility in the research. Potential participants will be screened using the Mini-Mental Status Examination (MMSE) [33] and exercise-related questionnaire [34].

Eligible participants are additionally required to satisfy the following criteria: (1) 60–80 years of age, (2) apparently healthy, free of cardiovascular disease, musculoskeletal problems, and psychiatric and neurological disorders, (3) normal body weight (body mass index is less than 25.0 and more than 18.5), (4) strong right handedness, (5) corrected visual acuity of at least 20/40, (6) a sedentary lifestyle, exercise at irregular base, and were assessed at the inactivity or low activity level, (7) voluntary participation in the study and willing to give written informed consent.

Participants will be excluded for any of the following: (1) objective cognitive impairment as measured by a MMSE score less than 25, (2) had experience playing table tennis in the past, (3) participated in similar research before, (4) any conflict with the objectives of this study.

### 2.3. Sample Size

The required sample size of the study was assessed by a priori power analysis (performed by G-Power). To attain a power of 0.8 (1-βerror probability), with a significant level at 0.05, 136 participants will be needed in the study.

Because the subjects are retired, older adults and due to the voluntary nature of enrollment, the attrition rate of the study is anticipated to be relatively low. When also considering other factors (such as health status, environment, and other emergency situations), a general attrition rate of 15% will be calculated in the recruitment plan task. Twenty-four additional participants will be recruited to compensate for the predicted drop-out rate, with 40 participants in each intervention condition.

To maximize compliance, the attendance and exercise performance of participants in the intervention will be monitored routinely throughout the trial.

### 2.4. Randomization

After signing the written informed consent, eligible participants will be randomly assigned to the active control group, and one of the three experimental intervention groups by the principal researchers. The randomization procedure will be run by an independent statistician using a computerized randomization program. The participants do not know the real purpose of the study, they are just told that they are participating in a project that aims to enrich the life of community dwelling older adults and that they are randomly allocated to different groups of the project according to a computerized randomization program. Participants and two principal researchers know the group assignments. Those who perform the intervention and outcome measures will be blinded to the allocation of participants.

### 2.5. Interventions and Control Condition

Participants in each intervention group will be told to enroll in a 24-week program and complete 50 min sessions three times a week, and to keep their regular daily life in normal living condition. All interventions will be conducted in the community center under the supervision of personnel who will be trained by the principle researchers to ensure the standardized administration of the intervention protocols.

The intervention conditions include: (1) open-skill exercise condition, which involves a combination of physical and cognitive training, (2) closed-skill exercise condition, which involves physical exercise training alone, (3) mobile game playing condition, which is considered as cognitive training alone, (4) active control condition.

#### 2.5.1. Open-Skill Exercise Condition

The open-skill exercise group will participate in a specially designed table tennis training program. The exercise intervention will be performed in a sequence of increasing complexity under the instruction of trainers. The trainers will be chosen from college students majoring in table tennis training.

The procedure of each single intervention session includes a warm-up, training in the basic skills of table tennis, playing table tennis with each other, special tasks designed to increase the cognitive load, and a cool down period. The aim of basic skills training is to improve general skills. It includes the following component: serving, backhand blocking, forehand attacking, left side blocking, and right side attacking (including footwork). After the coaches have demonstrated the basic skills, participants will practice with each other (or with the coaches). Training for each skill begins with simple movement, and then progresses to more complex skills, according to their degree of mastery.

To add more cognitive load to physical exercises during the process of learning the basic skills, tasks must be planned and a program of movement will be designed. This approach will be structured to achieve a particular type of training that is expected to relate to executive function. Table-tennis balls will be successively thrown to participant by a serving machine on one side of the ping-pong table. Participants stand on the other side of the table, and must anticipant the ball trajectory to intercept it. Balls are of two different colors; half are white and half are yellow. Three different tasks were designed to increase cognitive load. Participants must intercept the white ball but ignore the yellow ball in task 1, which requires inhibition function during movement. Task 2 requires participants to hit the white ball to the left side and the yellow ball to the right side across the ping-pong table. which needs task-switch ability. Task 3 was designed to train the working memory ability by asking participants to hit the ball to the left side if the previous ball is white and to the right if previous ball is yellow.

All participants will be required to move continuously during the training to maintain an average individual target heart rate reserve (HRR) of 40–59%.

#### 2.5.2. Closed-Skill Exercise Condition

The closed-skill exercise group will use a cycle ergometer or motor-driven treadmill with a moderate intensity (40–59% HRR). Each training session will include a warm-up, cycle ergometer riding or brisk walking/jogging for 30 min, and cool down period. Intensity and duration will be progressive during the training period. The Borg Rating of Perceived Exertion (RPE) will be used every 5 min during exercise to monitor the subjective feeling of exertion.

#### 2.5.3. Mobile Game Playing Condition

The cognitive training group will play a table tennis mobile game on the mobile phone without body movement. The mobile game is called Table Tennis Touch, and is a free sport action mobile multiplayer game featuring stunning graphics, intuitive swipe controls, high speed gameplay and multiple game modes developed by the British Indie Studio Yakuto. It offers a fantastic simulation of real table tennis playing settings. Players can serve, spin, and smash their way to winning. The older adults will be trained the first time, and will subsequently play under the supervision of the trainers.

#### 2.5.4. Active Control Condition

To control for the Hawthorne effect, which means participants may change their behavior just by being aware that they are being watched, the control group will also do some stretching and toning of the same duration and frequency as the intervention group. Social engagement will be similar among all groups.

### 2.6. Outcome Measures

Demographic variables will be collected during the evaluation, and include age, gender, weight, height, mental status, visual acuity, resting heart rate and blood pressure. The outcomes include objective and subjective measurements of cognitive and physical activities. All outcomes will be measured prior and immediately after the intervention, and at 3 months after the intervention. After 12 weeks, there will be an intermediate measurement, consisting of the primary outcome measures.

#### 2.6.1. Primary Outcome

The primary outcome measure of this study is objective visuospatial working memory. It is measured by the visuospatial working memory task (VWMT) developed by Guo et al., (2016), which includes both the passive storage and active manipulation of visuospatial information [32]. Participants will be asked to hold a 4 × 4 matrix with four solid black squares as a probe stimulus in memory for 3 s, and mentally rotate the matrix 90° to the right or the left at the same time. The task is used to decide whether the testing stimulus is consistent with the mentally rotated probe stimulus 90° either to the left or right.

Passive storage and active manipulation will be measured separately to reveal the mechanism driving differences on visuospatial working memory. The passive storage of visuospatial information will be tested by visuospatial short-term memory task (VSMT). The experimental paradigm of VSMT is similar to that of VWMT, except that subjects are instructed to retain the memory stimulus during the retention interval. Participants must determine whether the testing stimulus is identical to the memory stimulus. Active manipulation of visuospatial information will be tested by the visuospatial mental rotation task (VMTT). A pair of matrices will be presented on the screen for 6 sec. Participants must compare the two matrices to determine whether the one on the right side corresponds to a 90° rotation of the one on the left side.

#### 2.6.2. Secondary Outcomes

The subjective memory complaint (SMC) occurs long before the decline of working memory during aging [35]. It will also be measured in this study. SMC will be rated using the Memory Complaint Questionnaire (MAC-Q). The six-item MAC-Q requires participants to compare current memory function to memory function at earlier ages in daily scenarios. For each question, there are five possible answers, ranging from “much better now” (scored 5) to “much worse now” (scored 1). Serious SMC is defined as a total score below 15 [36]. MAC-Q demonstrates good internal consistency together with satisfactory test-retest reliability [37].

The attention network test (ANT) will be used to assess the alerting, orienting, and executive networks of the attentional system independently in a single test [38]. The test is a combination of the Flanker paradigm and a cueing task, with four cue types (no, central, double, and spatial) and three flanker types (congruent, incongruent, and neutral). The cue stimuli are asterisks, and are always valid to provide temporal information about the coming target stimuli. The target stimuli are five horizontal arrows that serve as flankers. Participants will be instructed to respond to the central arrow as fast and as accurately as possible [39,40].

A matrix reasoning test (MRT) will be used to measures abstract nonverbal reasoning ability. This test includes 35 items, in which the individual is required to look at an incomplete matrix and select the missing portion from five response options. This test is a subtest of the Wechsler Abbreviated Scale of Intelligence, and is relatively language free [41]. The score is generated based on the number of items completed correctly.

The Taiwan version of the International Physical Activity Questionnaire (IPAQ) will be used as a subjective measurement to estimate the total amount of physical activity in the preceding week.

The Mi Band wristband will be used to measure objective physical activity. Participants will be asked to wear the wristband all day from one week before the intervention, during the whole intervention, during the 3-month post-intervention. The wristband inserts a 3-axis acceleration sensor ADXL362 that measures the number of walking steps and distance travelled in everyday life. The wristband will also monitor the heart rate during intervention.

### 2.7. Data Collection, Management and Statistical Analysis

The outcome measures will be carried out conducted in the community center by trained research assistants who are not involved in the intervention. All outcome data will be collected and stored electronically. A data management team will be established to monitor the process of data collection, record and analysis once a week during the project. The data of the participants who withdraw will be treated as having no change from baseline after dropping out. Analysis of variance (ANOVA) or the χ^2^-test will be used to assess differences across groups with respect to age, percentage of females, BMI, scores on MMSE, visual acuity, resting heart rate, and blood pressure. If group differences are observed at baseline, these variables will be included as covariates in further analysis. Repeated-measures analysis of variance will be used for the primary and sedentary outcome measures. Dependent variables include the accuracy and reaction times on VWMT, VSMT and VMTT, the scores on alerting, orienting, and executive networks of ANT, Matrix reasoning test, MAC-Q and IPAQ, and the number of steps and distance. Independent variables include group (open-skill exercise, closed-skill exercise, cognitive training and control group) as the between subject factor, and the test time points (pre-test, intermediate-test, post-test, and 3-month follow-up test) as the within subject factor. To reveal the relationship between subjective memory complaint and objective working memory ability, Pearson correlation will be adopted to compute the correlation coefficient between the accuracy of VWMT and the MAC-Q score at different test time points.

## 3. Discussion

Previous studies that evaluated how open/closed-skill exercises affected the cognitive functions of older people mainly used a cross-sectional design [25,32]. There has been only one longitudinal study using a 6-month randomized controlled trial [27]. The longitudinal study showed that older adults who participated in open-skill exercises had better executive function compared to those that participated in closed-skill exercises. However, the mechanism underlying the additional advantage of open-skill exercises remains subject to debate. One major strength of the proposed study protocol is its design with four groups. In addition to the open-skill, closed-skill, and control group, there will be a cognitive training group. The inclusion of a cognitive training group provides the opportunity to separate the physical activity component and cognitive activity component by comparing differences among the groups. This approach might reveal the possible mechanisms driving the additional cognitive benefits of open-skill exercises. The closed-skill group and cognitive training group will only have the physical activity and cognitive activity component, respectively, while the open-skill exercise group will involve both physical and cognitive components. The cognitive training group will use a table tennis mobile game that differs previous computerized cognitive training [42,43], as it will utilize the same cognitive component as the open-skill exercise group.

Several studies have explored various methods to combine cognitive and physical training as a tentative approach to improve cognitive functions. On one hand, according to the cardiovascular fitness hypothesis, participant in chronic exercises may lead to changes in cardiovascular system, this change can increase cerebral blood flow and the up-regulation of neurotransmitters [44]. On the other hand, cognitive training can guide and integrate new neurons and synapses into existing neural networks [45]. Therefore, the combination of the two interventions may induce superimposed cognitive benefits. The combination of cognitive and physical training typically being completed separately or simultaneously. It has been demonstrated that simultaneous combination has a larger effect on cognitive function compared to their being conducted separately. The simultaneous combination is usually conducted as dual-tasks, such as pedaling on a stationary bike while learning the memory strategies and practicing the memory drills presented by a trainer on the screen in front of them [46]. These combinations are not popular by the elderly population due to the difficulty of implementation. Thus, open-skill exercises that are played in a dynamically changing, unpredictable, and external environment, and require a cognitive component, represent better choice. Similar to table tennis used in this proposed study, Moreau et al., (2015) designed training that was loosely based on freestyle wrestling. This approach favored new motor coordination via demonstrations and active trial-and-error problem-solving with a partner. After 8 weeks training, the designed sport group showed the largest gains in all cognitive measures. [47]. Open-skill exercises can be viewed as high ecologically validated approaches that combine cognitive and physical training simultaneously.

This proposed project differs to previous studies by measuring visuospatial working memory as a primary outcome, rather than general cognitive function. This is key because visuospatial working memory ability declines more severely after the age of 60. Additionally, a previous cross-sectional study of older adults only revealed a possible relationship between exercise mode and visuospatial working memory. In comparison, the proposed longitudinal project will explore the reasons why open-skill exercises generate more benefits on visuospatial working memory. The subjective memory complaint (SMC) will also be measured alongside the objective measures of working memory. SMC is defined as the reporting of memory problems without pathological results on neuropsychological tests. SMC is associated with later cognitive decline and a higher incidence of dementia [48]. A randomized controlled study reported that 12 weeks of exercise interventions induced beneficial effects on memory complaints (tested by MAC-Q), but not on objective memory performance (tested by the Prose Test and Rey test) [49]. In addition to SMC, the proposed study will investigate the association between MAC and visuospatial working memory, which have been subject to controversy in previous studies.

A limitation of the study is that outcome measures mainly include cognitive domains. However, the general aim of exercise is to provide benefits on the overall aspects of well-being, including both cognitive functions and physical performance. Apart from the included measures of physical activities, other measures of cardiovascular fitness and physical performance such as mobility, balance, strength and endurance will also be added to evaluate the physical functions in future studies.

## 4. Conclusions

The overall goal of this study is to compare the effects of open-skill exercises on cognition in older adults. If successful, the results of the proposed study provide important evidence-based recommendations for older adults to select the most efficient and effective exercise mode to improve cognitive function. In particular, this study will reveal the mechanism driving the additional cognitive benefits of open-skill exercises compared to closed-skill exercises.

## Figures and Tables

**Figure 1 brainsci-11-00609-f001:**
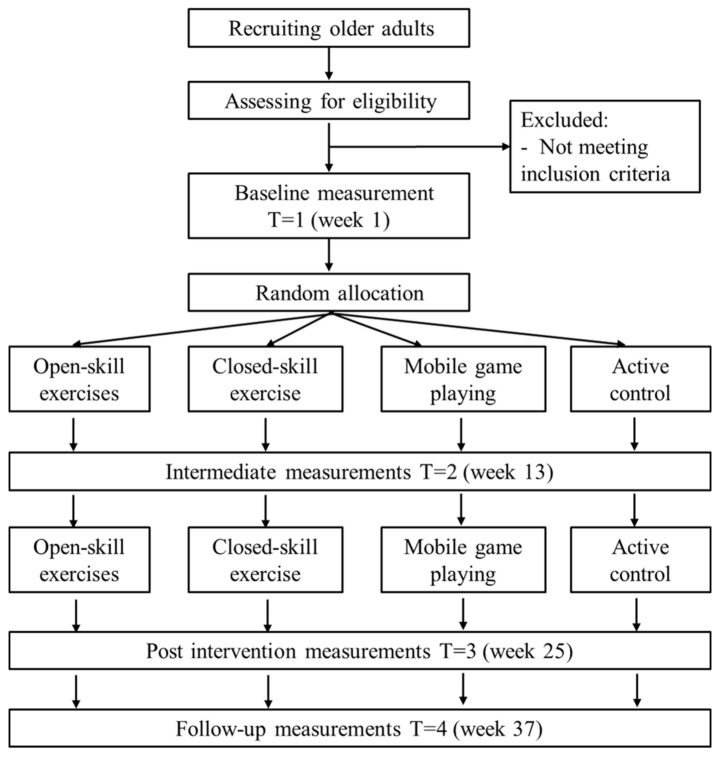
Flowchart of the study design.

**Figure 2 brainsci-11-00609-f002:**
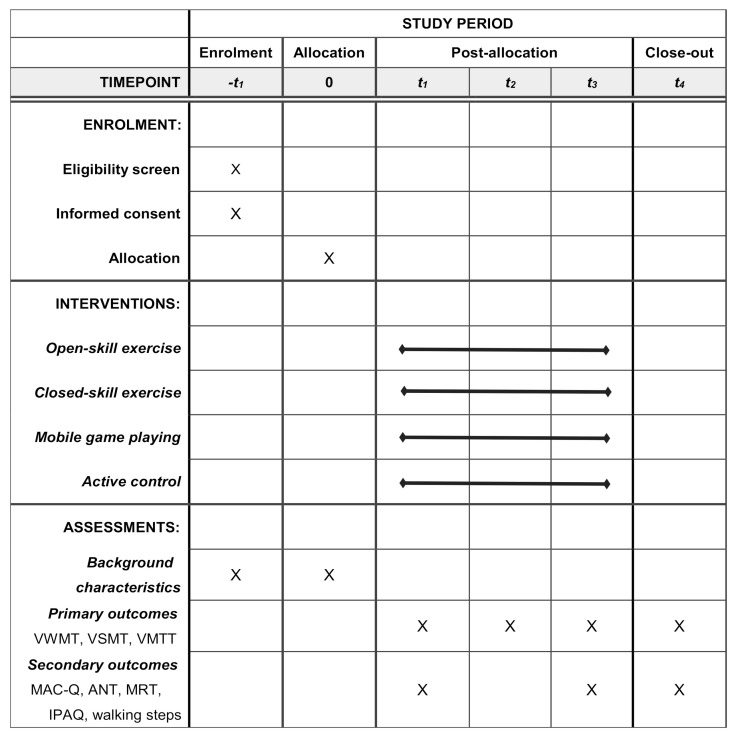
Schedule of enrolment, interventions, and assessments of the trial. t_1_: baseline (week 1), t_2_: intermediate of intervention (week 13), t_3_: immediately post-intervention (week 25), t_4_: twelve weeks after intervention (week 37), VWMT: visuospatial working memory task, VSMT: visuospatial short-term memory task, VMTT: visuospatial mental rotation task, MAC-Q: Memory Complaint Questionnaire, ANT: attention network test, MRT: matrix reasoning test, IPAQ: International Physical Activity Questionnaire.

## Data Availability

The data are not publicly available, as they have not yet been collected. The data will be uploaded at the Chinese Clinical Trail Registry site when finishing the study.

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
