# Peer review of "Effects of Open-Skill Exercises on Cognition on Community Dwelling Older Adults: Protocol of a Randomized Controlled Trial"

_brainsci, 2021, doi:10.3390/brainsci11050609_

Round 1

Reviewer 1 Report

This study protocol aimed to investigate the effect of open-skill exercise training on cognitive performance in community dwelling older adults compared with closed-skill exercise, cognitive training, and active control. The manuscript is well written, easy to read, and covers a topic in the area of exercise-cognition relationship that is worth being investigated. The protocol is complete and well structured. I would congratulate the Authors for their work. I have only a couple of minor suggestion to improve the completeness of the protocol.

  • The study will be on adults. However, to provide a clear overview of the type of exercise-cognition relationship, I do believe that the cited literature should include also studies on children, as this topic is currently being investigated also on children. Here some recent references to cite:

Becker, D.R., McClelland, M.M., Geldhof, G.J., Gunter, K.B., MacDonald, M., 2018. Open-Skilled Sport, Sport Intensity, Executive Function, and Academic Achievement in Grade School Children. Early Educ. Dev. 29, 939–955. https://doi.org/10.1080/10409289.2018.1479079

Formenti, D., Trecroci, A., Duca, M., Cavaggioni, L., D’Angelo, F., Passi, A., Longo, S., Alberti, G., 2021. Differences in inhibitory control and motor fitness in children practicing open and closed skill sports. Sci. Rep. 11, 4033. https://doi.org/10.1038/s41598-021-82698-z

Gu, Q., Zou, L., Loprinzi, P.D., Quan, M., Huang, T., 2019. Effects of Open Versus Closed Skill Exercise on Cognitive Function: A Systematic Review. Front. Psychol. 10. https://doi.org/10.3389/fpsyg.2019.01707

  • The primary outcome of this study is cognitive performance. However, the general aim of exercise is to provide benefits on the overall aspects of well-being (both cognitive functions and physical performance). Apart from the included measures of physical activity (and not physical functions, please correct) such as IPAQ and step-counts based on accelerometer, I would suggest to include also some measures of physical performance. For example, there are simple and easy tests to estimate aerobic capacity and physical functions. In any case, if not included, I believe that this notion should be added at the end of the Discussion to take into consideration in further studies.

Reviewer 2 Report

I welcome studies that introduce novelty and applicability on a topic of older adults and cognition.  In fact, I am open to be persuaded to deep understand the relationship between the effec of open-skill exerciste training on cognitive function compared with closed-skill exercise/cognitive training. Hence, I am some sympathy with the author's intentions but the paper does not make clear the perspective that the author expresses.

On the one hand, I emphasize the importance of the study and of the sample collected. It is a really complicated to access such high-quality of sample (older people. In addition, is also to be apreciated the constructive criticism that you show in the article, but i think that serious issues requiring urgents solutions and wich need to be remedied. 

On the other hand, it is not clear the introduction part and is very poor the final discussions. I feel that the practical implications and limitations must be further developed and receive a deep explain.

In addition, you should discuss your study emphasized on cardiovascular hypothesis (chronic exercise) and arousal theory or hipofrontality theory for acute exercise. Is very important in dual task perspective.

Where is the results part? is missing

I considerate that conclusions and discusion can still be significantly improved.

Round 2

Reviewer 2 Report

Thank you very much, particularly for the discussion, limitations and conclusions. I think that your manuscript is stronger as a result of these modifications.

This manuscript is a resubmission of an earlier submission. The following is a list of the peer review reports and author responses from that submission.